

# Anti-atherosclerotic vaccination against *Porphyromonas gingivalis* as a potential comparator of statin in mice

Hyun-Su Ha[1], Tae Young Kim[1], Soo Jung Han[2], Hak-Joon Sung[1], Kyoung Yul Seo[3] and Jong-Won Ha[4]

[1] Department of Medical Engineering, Yonsei University College of Medicine, Seoul, Republic of Korea
[2] The Institute of Vision Research, Department of Ophthalmology, Yonsei University College of Medicine, Seoul, Republic of Korea
[3] Department of Ophthalmology, Yonsei University College of Medicine, Seoul, Republic of Korea
[4] Cardiology Division, Severance Hospital, Yonsei University College of Medicine, Seoul, Republic of Korea

## ABSTRACT

**Background**. *Porphyromonas gingivalis (Pg)* is an oral anaerobe which damages teeth and periodontal tissues. Its body infection is known to cause chronic inflammation, thereby inducing an early stage of atherosclerosis through humoral immune actions. Hence, vaccination by immunizing the proteins of *P. gingivalis (Pg)* post sonication with heating may prevent atherosclerosis. This study aimed to compare the effect of its vaccination with statin, which effectively prevents atherosclerosis by lowering lipids.

**Methods**. The vaccine was produced by sonicating *P. gingivalis* through heating, and a total of 32 male APOE-/-mice (8-week old) were subjected Western diet for 8 weeks, in order to induce atherosclerosis in a physiological manner. Then, the mice were grouped to undergo four treatment conditions (i.e., no treatment, pitavastatin, vaccine, or pitavastatin with vaccine). Vaccination was conducted through nasal immunization and confirmed by a Pg-specific humoral immune reaction. Then, half of the mice in each group were orally injected with *P. gingivalis* for the next 5 weeks while the other half remained uninfected, generating a total of eight groups ($n = 4$/group). The mice were sacrificed at 3 weeks after the last injection. After harvesting the aorta, Oil Red O staining of en face was conducted with imaging and image analysis, and plaque formation was quantitatively determined.

**Results**. Compared to no treatment, the vaccination through nasal immunization significantly reduced the atherosclerotic plaque sizes in APOE -/- mice under Western diet to the comparable level of statin group. When both vaccine and statin were used, no clear synergistic effect was observed as opposed to expectation.

**Conclusions**. This study revealed that nasal immunization of heat shock *P. gingivalis* has a significant impact on the prevention of arteriosclerosis and acts as a potential comparator of statin.

Corresponding author
Jong-Won Ha, jwha@yuhs.ac

## INTRODUCTION

Atherosclerosis is an apolipoprotein-mediated immune-inflammatory disease that acts as a major cause of cardiovascular diseases. A passive build-up of cholesterol was known to promote atheroma formation, thereby inducing vascular stenosis. A series of recent studies have reported that recruiting inflammatory and immune cells and their pathological actions with phenotype changes are major drivers of the atherosclerotic progress (*Libby, Lichtman & Hansson, 2013*).

Periodontitis is a chronic inflammatory disease that is seen most often in the mouth, leading to atherosclerosis (*Slavicek et al., 2009*). In the 1960s, it was first observed that periodontitis patients were accompanied with atherosclerosis of gingival vascular bed. Then, it has been found that a gram-negative anaerobe, *Porphyromonas gingivalis,* is not only a major actor to induce periodontitis but also has potential to induce atherosclerosis. (*Quintarelli, 1957*; *Stahl, Witkin & Scopp, 1962*). Since then, studies on relevant clinical and pre-clinical models have revealed a clear causative role of *P. gingivalis* in inducing atherosclerosis (*Kramer et al., 2017*), and mechanistic relationships of periodontitis with systemic diseases have become a popular subject in current studies (*Konkel, O'Boyle & Krishnan, 2019*). As a result, studies have reported that innate and adaptive immune systems are involved in operating the mechanism, and thus suggested a potential protective effect of vaccination by sonicating *P. gingivalis* through heating (*Choi et al., 2011*).

Here, we investigated a causative action of *P. gingivalis* in inducing atherosclerosis using ApoE knock-out mice under Western diet as a pathophysiological model. A *P. gingivalis* solution was subjected to heat shock through sonication, thereby generating vaccine. This vaccine was injected to mice by nasal immunization, and the formation of Pg-specific antibody was checked by immunoassay. The therapeutic effect of vaccine to prevent atherosclerosis was determined in the mice through comparison with pitavastatin.

## MATERIALS & METHODS

### *P. gingivalis* and pitavastatin

*P. gingivalis* strain ATCC 53978 was cultured with tryptic soy broth containing haemin and vitamin K (Becton Dickinson, Tokyo Chemical Industry Co., Ltd, Sigma-Aldrich) in an anaerobic chamber with 80% N2, 10% H2, and 10% CO2. The bacteria concentration was standardized using a spectrophotometer at OD 650 nm (0.1 of *P. gingivalis* = $10^{10}$ bacteria/ml). Vaccine was produced by heat-shocking of *P. gingivalis* solution through sonicating three times for 15 s each. Pitavastatin was dissolved in DMSO with PBS dilution (1:1 ratio) by sonicating for 1 min to improve the solubility. A total of $10^9$ CFUs of live *P. gingivalis* were suspended in 100 μL PBS with 2% carboxymethyl cellulose (Sigma-Aldrich) for animal studies.

### Mouse and diet

Experimental procedures, including housing and care, were conducted according to the regulation of the National Institute of Health and approved by the Ethics Committee and the Institutional Animal Care and Use Committee of Yonsei University, College of

Medicine (IACUC Approval number, 2017-0167). Eight-week-old ApoE −/− male mice were purchased from Jackson Laboratories, and housed at 23 ∼24 °C in a 12-hour light/dark cycle. APOE -/- mice were fed by 21gm% Western diet powder (D12079B, Research Diet) for 8 weeks and monitored daily. The power analysis was used to calculate the minimum number N at a significance level of 0.01 and power of 0.8. The minimum number N was 4.245 ($n = 4$).

### Pre-check of vaccination in mice

Mice were nasally immunized with 10 ug of vaccine with 1 ug of Cholera toxin every 2 weeks for 5 weeks, from week 11 to week 15. Serum and mucosal wash samples were collected at week 16 following a previous study (*Namikoshi et al., 2003*). Serum was obtained through a tail vein, and tear-wash samples were obtained by lavaging with 10 μl of PBS per eye. Saliva was obtained by intraperitoneal injection of pilocarpine (500 mg/kg body weight; Sigma-Aldrich) into the mice. Fecal extracts were obtained by dissolving the feces in PBS containing 0.1% sodium azide. The feces were mixed by vortexing and centrifuged, and the supernatants were collected for assays.

Successful vaccination was determined using the collected body samples in PBS by reacting IgG and IgA in ELISA plates (Falcon, Franklin Lakes, NJ) overnight at 4 °C. ELISA plate coated with sonicated whole cell extracts of *P. gingivalis* ATCC 53978(10 μg/ml) were blocked with 1% BSA (Sigma-Aldrich) in PBS. Samples were then reacted with HRP-conjugated goat anti-mouse IgG or IgA Ab (Southern Biotechnology Associates) in each well, followed by incubation overnight at 4 °C. Color signals were generated by adding tetramethylbenzidine solution (Life Technologies) until the addition of stopping solution (0.5 N HCl). Then, the signals were measured at 450 nm on an ELISA reader (Molecular Devices). Endpoint titers of Ag-specific Ab were expressed as reciprocal log2 titers of the last dilution that showed >0.1 absorbance over the background levels.

### Bacterial challenging of mice with anti-atherosclerotic effects

A total of 32 male APOE-/-mice (8-week old) were subjected to Western diet for 8 weeks, in order to induce atherosclerosis in a physiological manner (Fig. 1). After the period of adaptation, the mice were randomly and equally grouped to undergo four treatment conditions (i.e., no treatment, pitavastatin, vaccine, or pitavastatin with vaccine). The vaccination was conducted through nasal immunization and confirmed by a Pg-specific humoral immune reaction. Then, half of the mice in each group were orally injected with *P. gingivalis* for the next 5 weeks while the other half remained uninfected, generating a total of eight groups ($n = 4$/group). Pitavastatin (30 mg/kg/week) was injected three times (10 mg/mL/injection) every week for 5 weeks. *P. gingivalis* was administrated through the oral route five times per week for 5 weeks at a dose of $10^9$ CFU. The mice were euthanized with carbon dioxide gas, and the aortas were harvested for analysis. After cleaning the adventitia, the aorta was cut open longitudinally under a dissecting microscope to be analyzed entirely. The en face aortas were imaged by Oil Red O staining and HP Printer scanning. According to the method of a previous study (*Lin et al., 2015*), the images were quantitatively analyzed to determine the size of atherosclerotic plaque using Image J software (NIH, MD, US).

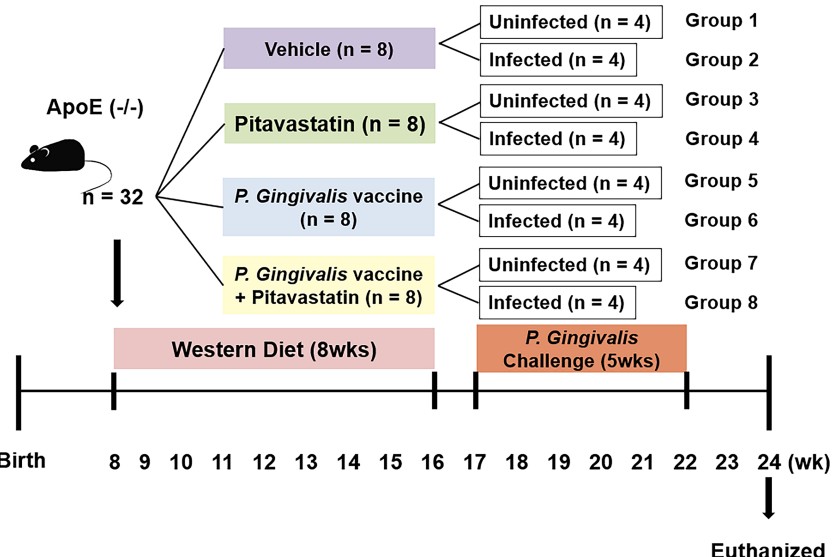

**Figure 1 Animal groups and experiment schedule.** Eight-week-old male ApoE−/−mice were fed with western diet powder for 8wks. Mice were then grouped to undergo four treatment conditions (i.e., no treatment, pitavastatin, vaccine, or pitavastatin with vaccine). The half number of mice in each group was orally injected with *P.gingivalis* for the next five weeks while the other half remained uninfected, generating a total of eight groups ($n = 4$/group).

When damage occurred during the process of staining and analyzing the tissue, it was excluded from the results; and a total of seven animals were excluded.

## Statistical analyses
Analyses were performed using SPSS 23.0 (Systat software, Chicago, IL). One-way ANOVA with Tukey multiple comparisons test was performed to compare the groups. *P* values less than 0.05 were considered significant.

# RESULTS

## Antibody reactions to body samples as working validation of vaccination
The atherosclerosis model was produced by subjecting ApoE-/- male to either normal or Western diet (WD), and the formation of atherosclerotic plaques was determined by Oil Red O staining with quantitative image analysis. The normal diet group started to show plaque formation at 20 weeks in contrast to 4 weeks in the WD group, and the plaque sizes of both groups grew over time. The WD production (D12079B) was used by refereeing the previous studies to profile microbiome in the gut (*Chan et al., 2016*) and to analyze atherosclerotic burden in the disease models using the WD product (*Hutter et al., 2015*).

Successful vaccination was determined based on antigen formation of the collected body samples. ELISA assays were used to determine IgG reaction to serum as well as IgA reaction to fecal, saliva, and tear. The nasal immunization of mice with vaccine and cholera toxin with or without PTV resulted in successful formation of antigens in the corresponding

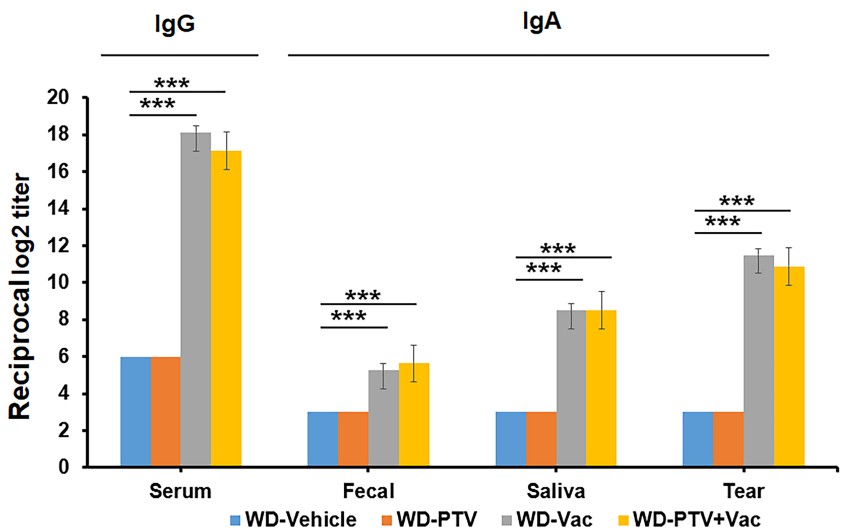

**Figure 2** **Pre-check of vaccination by immune response.** IgG reaction to serum as well as IgA reaction to fecal, saliva, and tear were determined by ELISA in mice under western diet post pitavastatin (PTV) administration and/or vaccination (Vac) ($n = 8$). PBS injection served as a vehicle control. Among the test groups, successful vaccination by both Vac and PTV+Vac was indicated significant increases in the expression of IgG and IgA. $p$-value: * < 0.05, ** < 0.01, *** < 0.001.

samples compared to the groups without vaccination (Fig. 2). These results indicated that the components and methods are effective in approaching vaccination.

## Anti-atherosclerotic effect of vaccine in comparison with PTV

The effect of vaccination to prevent atherosclerosis against repeated infection of *P. gingivalis* was compared with the treatment and co-treatment of PTV (Fig. 3). Atherosclerosis was induced in APOE -/- mice in physiological fashion by 8-week Western diet, and vaccination and PTV treatment were conducted prior to *P. gingivalis*. The infected mice with no treatment served as a control (Group 2) for comparison with the other groups, since this control represented the model of oral infection by *P. gingivalis* in a causative relation with the atherosclerosis progress. The effective prevention by vaccination (Group 6) against *P. gingivalis* infection was suggested by Oil Red O staining (Fig. 3A) with quantification of atherosclerotic plaques (Fig. 3B) in the en face of each aorta. Regarding the quantitative image analysis of the lesion area (% of total area) after Oil Red O staining, Group 1 (*Pg*-uninfected with no treatment) exhibited 37.07 ± 2.42% out of the total area compared to 92.64 ± 16.71% of Group 2 (*Pg*-infected mice with no treatment). This result confirmed that the disease model of infection was successfully produced by Western diet and *Pg* infection, as indicated by the more than two-fold difference in the formation of atherosclerotic plaques between the two groups.

Group 6 (vaccination) and Group 4 (PTV treatment) showed comparable anti-atherosclerotic effects, as shown by 16.18 ± 1.52% and 19.59 ± 1.83% out of the total area, respectively. However, the co-treatment group (Group 8), with 15.72 ± 2.82% (of the total area), did not show significant synergistic effects compared to our expectation.

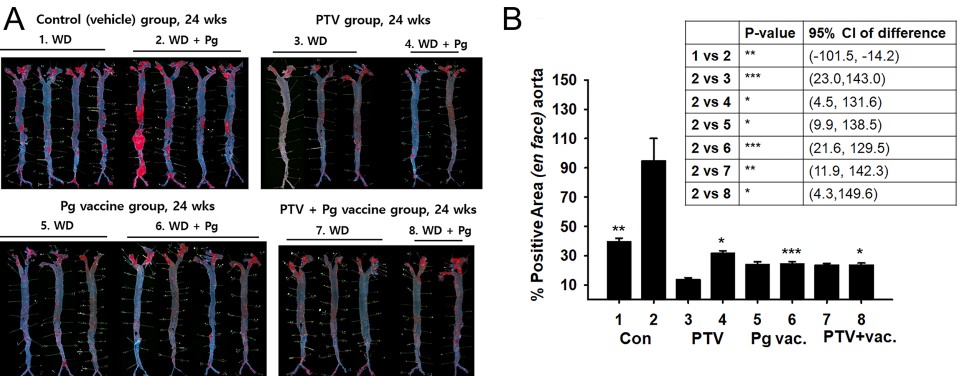

**Figure 3  Anti-atherosclerosis effect of vaccination and PTV against *P. gingivalis* infection.** (A) Oil-red O staining in the en face of each aorta (B) quantification of atherosclerotic plaques indicates the athero-protective effect of vaccination (Vac) and/or PTV treatment. PBS injection served as a vehicle control. *p*-value: * < 0.05, ** < 0.01, *** < 0.001.

The uninfected groups did not exhibit meaningful differences in the response, indicating a key role of *P. gingivalis* infection in establishing the disease model.

## DISCUSSION

Periodontal disease damages surrounding tissues over time, as the colonization of gram (-) bacteria around a gingival tissue leads to chronic inflammation. Due to the mechanistic link to chronic inflammation, this disease acts as a risk factor to induce cardiovascular diseases, such as early stage atherosclerosis (*Miyajima et al., 2014*) and myocardial infarction stroke (*Lockhart et al., 2012*). As a mechanism for periodontitis to induce atherosclerosis, it is suggested that the heat shock protein (HSP) of *P. gingivalis* serves as an antigen peptide pathogen that provides an epitope in promoting atherosclerotic autoimmunity, thereby activating T and B cells by triggering inflammatory responses during early stage atherosclerosis (*Choi et al., 2011*). Moreover, a previous study has reported that *P. gingivalis* invades into the vascular endothelium and vessel wall, resulting in activation of atheroma formation in the site (*Velsko et al., 2014*).

Also, a study has supported the report about the pathogen action of *P. gingivalis* by elucidating that a high titer of Anti-HSP60 Ab regulated the development and progress of atherosclerosis, cardiovascular disease, and cerebral infarction (*Mandal, Jahangiri & Xu, 2004*). Following this effort, another study produced GroEL as an anti-atherogenic vaccine using a recombinant HSP 60 from *P. gingivalis* (*Hagiwara et al., 2014*). Its sublingual immunization suppressed vascular lesion formation by reducing the expressions of CRP, MCP-1, and ox-LDL; therefore, this vaccine was suggested as a promising therapeutic factor to suppress *P. gingivalis*-mediated atherosclerosis.

Yukiko et al. found that nasal immunization of ApoE knock-out mice with 40 kDa outer membrane protein (OMP) of *P. gingivalis* reduced relevant atherosclerosis symptoms (*Koizumi et al., 2009*), Furthermore, Takashi et al. used CpG oligodeoxynucleotides (ODN) as an adjunctive together with 40kDa OMP (*Takeuchi, Hashizume-Takizawa & Kobayashi, 2017*). Their study reported that when this combination was applied for oral vaccination, IgG response and atherosclerotic plaque formation significantly attenuated. Joo et al. also found that nasal immunization of HSP 60-derived peptide 14 prolonged atherosclerotic progress by promoting IFN- γ release with reduction of Th17-mediated immunity (*Joo et al., 2020*). These previous studies are clearly aligned with our results regarding the proper responses of IgG and IgA by nasal immunization of vaccine.

Statin is a type of HMG-CoA reductase which reduces the LDL-Cholesterol level, thereby attenuating the risk of atherosclerotic cardiovascular disease. When Suh et al. injected rosuvastatin into mice with periodontal disease by ligature, atherosclerosis was effectively prevented (*Suh et al., 2020*). A novelty of our study lies in the fact that it is the first to investigate the effect of vaccination in comparison with statin, as well as the synergistic effect of their co-treatment. Our study also revealed that the infection of ApoE knock-out mice by *P. gingivalis* was induced efficiently in collaboration with Western diet, and the vaccination effect was comparable with that of pitavastatin in the prevention of atherosclerosis against the infection.

However, further studies are definitely required to apply the vaccine to reduce CVD risks in a larger animal model for clinical translation. There are also limitations to address through future studies. First, the small scale of the animal experiments might not provide conclusive differences between the statin and vaccine groups. Moreover, the first aorta of WD+Pg group appeared to be more positive to staining compared to the other samples. Although the first sample was excluded, the results were not significantly changed. Nonetheless, since the location and degree of arteriosclerosis of mice were not constant, more experimental groups may be required. Second, since some aorta samples were damaged during preparation for staining, these samples were not included in the analysis. Regardless of such limitations, our study still adds value to the ongoing efforts by suggesting a potential for vaccination with regard to wide-spread oral infection.

## CONCLUSIONS

The vaccine was successfully produced in a form of proteins of *P. gingivalis* by sonication through heating, and vaccination was also effectively conducted through nasal immunization in mice. This vaccine exerted a promising effect in preventing atherosclerosis, and thus can be considered as a potential substitute of statin through further investigation in the future.

### Funding

This work was supported by the National Research Foundation of Korea (NRF) (No. 2016M3A9E9941743), and Bio & Medical Technology Development Program of the

National Research Foundation (NRF) funded by the Ministry of Science and ICT (2016M3A9E9941746), and this study was funded by the JW Pharmaceutical Corporation. The funders had no role in study design, data collection and analysis, decision to publish, or preparation of the manuscript.

## Grant Disclosures

The following grant information was disclosed by the authors:

National Research Foundation of Korea (NRF): 2016M3A9E9941743.

Bio & Medical Technology Development Program of the National Research Foundation (NRF).

Ministry of Science and ICT: 2016M3A9E9941746.

JW Pharmaceutical Corporation.

## Competing Interests

The authors declare there are no competing interests.

## Author Contributions

- Hyun-Su Ha conceived and designed the experiments, performed the experiments, analyzed the data, authored or reviewed drafts of the paper, and approved the final draft.
- Tae Young Kim performed the experiments, analyzed the data, prepared figures and/or tables, and approved the final draft.
- Soo Jung Han performed the experiments, prepared figures and/or tables, and approved the final draft.
- Hak-Joon Sung, Kyoung Yul Seo and Jong-Won Ha conceived and designed the experiments, authored or reviewed drafts of the paper, and approved the final draft.

## Animal Ethics

The following information was supplied relating to ethical approvals (i.e., approving body and any reference numbers):

All animal experiments and management procedures were approved by the Institutional Animal Care and Committee (IACUC) of the Yonsei Laboratory Animal Research -Center (YLARC) [Permit No. 2017-0167].

## Data Availability

The raw measurements are available in the Supplementary Files.

## Supplemental Information

Supplemental information for this article can be found online at http://dx.doi.org/10.7717/peerj.11293#supplemental-information.

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
