# Peer review of "Anti-atherosclerotic vaccination against Porphyromonas gingivalis as a potential comparator of statin in mice"

_PeerJ, doi:10.7717/peerj.11293_

## Round 0.1 · original submission · Major Revisions

Please address the critiques of all reviewers and revise the manuscript accordingly.

·

Basic reporting

This paper compared the effect of vaccinatino along vs. the effect of vaccination plus statin. The author's conclusion is that vaccination itself is enough to prevent atherosclerosis.
The reporting is clear, background information is adequate.

Experimental design

The author showed very clear experiment design in figure 1.

Validity of the findings

The results for 24 weeks is clear. The conclusion would be stronger if longer time data can be available to serve as another data point

·

Basic reporting

Unambiguous language is mostly used throughout the manuscript, through there are a few places that warrant revision:
- Line 50: "The vaccination was conducted by through...". Either delete the "by", or "through"
- Line 51: "Then, the half number of mice...". Can revise to: "Then, half the mice..."
- Line 122: "Saliva was obtained intraperitoneal...". Can revise to: "Salive was obtained by intraperitoneal..."
- Line 139: "...were subjected wester diet...". Can revise to: "...were subjected to western..."
- Line 144: "Then, the half number of mice...". Can revise to: "Then, half the number of mice..."
- Line 148: administrate can be revised to administered
- Line 228: "...p. gingivalis infection induced efficiently in collaboration with...". Can revise to: "...p. gingivalis was induced effectively by..."
- Line 233: "First, a small scale of animal experiments might affect the unclear difference...". Can revise to: "First, the small scale of the animal experiments might not provide conclusive..."
- Line 237: "wide-spreading" can be revised to "wide-spread"
- Line 243: "comparator" can be revised to "substitute"

Experimental design

Experiment is well designed, and research question is well defined, relevant and meaningful.

Validity of the findings

The result on the validation of vaccination is sound. The result and conclusion on the anti-artherosclerotic effect of vaccine, however, was not very strong. The largest factor is the result on WD+Pg. The first aorta has very large oil red staining, indicating large artherosclerotic plague, while the rest are very small. Without the result from the first aorta, the extent of plague from the rest of the aorta in the WD+Pg group will be small, and significance level of plague reduction from PTV or Pg vaccine group will be smaller. This limitation should be discussed in more detail in the discussion section.

Reviewer 3 ·

Basic reporting

Peridontitis is a chronic infectious disease caused by Porphyromonas gingivalis and it acts as an independent risk factor for cardiovascular disease by triggering the formation of atherosclerotic plaques. Innate and adaptive immune response to bacterial heat shock proteins is suggested to promote the progression of atherosclerosis. Several mice studies used recombinant proteins from P. gingivalis for vaccination. Here, Ha et al., vaccinated by immunizing proteins of P. gingivalis post sonication with heating. Ha et al., used n=32 ApoE (-/-) mice in their study grouping them into four different treatment conditions for vaccination. Ha et al., for the first time looked at efficacy of vaccination in reducing atherosclerotic plaques in the presence or absence of pitavastatin (PTV), a statin that are known to prevent atherosclerosis by lowering lipids. Ha et al., determined the immune response to vaccination by measuring IgG and IgA levels by ELISA. They found significantly higher levels of IgG and IgA levels compared to controls indicating vaccination is fruitful. Ha et al., showed that atherosclerotic plaque size is reduced due to vaccination with or without PTV treatment. Ha et al., conclude that presence of PTV had no synergistic or additive effect on atheroprotective effect of vaccination. This is altogether a nice paper with carefully acquired data. It is well written, and it provides an evidence that vaccination by using sonicated P. gingivalis with heating had an atheroprotective effect that is comparable to PTV treatment.

Experimental design

It is a original work with research question well defined. Methods section described with sufficient detail.

Validity of the findings

Comments:
1. The authors determined the HDL and LDL levels in all groups of animals in lipid analysis. I think it would be nice to plot the data and show it in the main figure and talk about why they don’t see any reduction in LDL levels in groups that are treated with vaccination in the presence or absence of PTV. For instance, in pitavastin treatment group, I expected the reduction of LDL-cholesterol level, but I don’t see any difference in their values compared to untreated animals fed with western diet. Good news is I see increase in HDL levels. The authors claim the vaccination is comparative to PTV treatment in reducing atherosclerotic plaques, but I don’t see any changes in both HDL and LDL cholesterol levels in groups vaccinated with P. gingivalis. So, they should discuss what could be the reason for reducing plaque area in vaccinated group in the presence or absence of PTV.
2. It would also be nice if authors can measure the cytokine production and CD4+CD25+ Tregs cells and determine if they play a role in reducing atherosclerotic plaque area similar to Joo et al., 2020.
3. In figure 2, please find a way to describe wd (western diet).

---

## Round 0.2 · Minor Revisions

Please address the remaining issues pointed by the reviewer.

·

Basic reporting

The author has addressed comments

Experimental design

The comments was addressed by the author

Validity of the findings

The comment was addressed

·

Basic reporting

Basic reporting is mostly clear and unambiguous. There are just a number of places that can benefit from grammatical corrections.

Experimental design

Experimental design is clear and well defined

Validity of the findings

My concerns have been addressed, and limitations are clearly written. Figure 3A seems to be missing. If the authors can add that, it will be great.

Additional comments

I think the paper is good to go with minor revision with the language and grammar.

Reviewer 3 ·

Basic reporting

Ha et al., have addressed most of the concerns raised by the reviewers. It is also understandable that they couldn't perform few animal experiments because of COVID issues. I am happy with their manuscript for publication in peerJ.

Experimental design

No comments

Validity of the findings

No comments.

---

## Round 0.3 · accepted · Accept

All remaining issues were adequately addressed and the manuscript was amended accordingly. Therefore, I am glad to accept revised manuscript.